# Quantum-circuit refrigerator

Kuan Yen Tan[1], Matti Partanen[1], Russell E. Lake[1,†], Joonas Govenius[1,†], Shumpei Masuda[1] & Mikko Möttönen[1]

Quantum technology promises revolutionizing applications in information processing, communications, sensing and modelling. However, efficient on-demand cooling of the functional quantum degrees of freedom remains challenging in many solid-state implementations, such as superconducting circuits. Here we demonstrate direct cooling of a superconducting resonator mode using voltage-controllable electron tunnelling in a nanoscale refrigerator. This result is revealed by a decreased electron temperature at a resonator-coupled probe resistor, even for an elevated electron temperature at the refrigerator. Our conclusions are verified by control experiments and by a good quantitative agreement between theory and experimental observations at various operation voltages and bath temperatures. In the future, we aim to remove spurious dissipation introduced by our refrigerator and to decrease the operational temperature. Such an ideal quantum-circuit refrigerator has potential applications in the initialization of quantum electric devices. In the superconducting quantum computer, for example, fast and accurate reset of the quantum memory is needed.

---

[1] QCD Labs, COMP Centre of Excellence, Department of Applied Physics, Aalto University, PO Box 13500, FI-00076 Aalto, Finland. † Present addresses: National Institute of Standards and Technology, Boulder, Colorado 80305, USA (R.E.L); Department of Physics, Stockholm University, AlbaNova University Center, SE-10691 Stockholm, Sweden (J.G). Correspondence and requests for materials should be addressed to K.Y.T. (email: kuan.tan@aalto.fi) or to M.M. (email: mikko.mottonen@aalto.fi).

Engineered quantum systems show great potential in providing a spectrum of devices superior to the present state of the art in information technological applications. Since quantum technological devices operate at the level of single energy quanta, they exhibit very low tolerance against external perturbations. Consequently, they need to be extremely well isolated from all sources of dissipation during their quantum coherent operation. These properties typically lead to an elevated steady-state temperature and long natural initialization times. Thus, finding a versatile active refrigerator for quantum devices is of great importance.

One of the greatest challenges of this century is to build a working large-scale quantum computer[1,2]. To date, a superconducting quantum computer[3] has reached the required gate and measurement accuracy thresholds for fault-tolerant quantum error correction[4,5] (see also ref. 6). This device builds on the decade-long development of circuit quantum electrodynamics[7–10], that is, the study of superconducting quantum bits, qubits, coupled to on-chip microwave resonators[11–13]. Although several methods have been demonstrated to initialize superconducting qubits[14–20] and resonators[21], they are typically suited only for a very specific type of a system and the achieved fidelities fall below the demanding requirements of efficient fault-tolerant quantum computing. Thus, circuit quantum electrodynamics provides an ideal context for the demonstration of a quantum refrigerator.

Electronic microcoolers based on normal metal–insulator–superconductor (NIS) tunnel junctions[22,23] offer opportunities[24] to cool electron systems well below the temperature of the phonon bath even at macroscopic sizes[25]. Due to the ideally exponential tunability of the NIS cooling and input powers using an applied bias voltage, these tunnel junctions are attractive candidates for quantum refrigerators. Although single-charge tunnelling has previously been demonstrated to emit and absorb energy quanta[26–29] even in applications such as the quantum cascade laser[30], and artificial-atom masers[31–33], it has not been experimentally utilized to directly cool engineered quantum circuits. Even the recently demonstrated autonomous Maxwell's demon has only been used to refrigerate dissipative electron systems[34].

In this work, we utilize photon-assisted electron tunnelling to cool a prototype superconducting quantum circuit, namely a transmission line resonator. The tunnelling takes place in NIS junctions. Since an electron tunnelling event changes the voltage across the resonator, it may simultaneously induce quantum transitions in the resonator modes[35]. Owing to the energy gap in the superconductor density of states, only the most energetic electrons can escape the normal metal in analogy with evaporative cooling. Thus, a tunnelling process where the electron absorbs a photon from the resonator, simultaneously increasing its energy, is favoured against photon emission. This is the key phenomenon leading to the refrigeration of the resonator. We refer to the NIS tunnel junctions and their coupling circuitry as the quantum-circuit refrigerator (QCR), which is a stand-alone component for cooling the operational quantum degrees of freedom in different types of under-damped quantum electric devices.

The photon-assisted cooling of the resonator mode is suggested by our qualitative observation that a distant probe resistor electrically coupled to the resonator cools down even if the temperature of the QCR, and of the other heat baths, is elevated. This claim is reinforced by the absence of cooling in a control sample, in which the coupling between the probe and the microwave resonator is suppressed. Furthermore, we obtain a good quantitative agreement between our theoretical model and the experimental results over a broad range of QCR operation

voltages and bath temperatures, providing firm evidence of our conclusion that we directly refrigerate the resonator mode. We also verify that the resonator has a well-defined resonance.

## Results

**Experimental samples.** Figure 1a–d shows the active sample where a QCR and a probe resistor are embedded near the opposite ends of a superconducting coplanar-waveguide resonator. The refrigerator involves a pair of NIS junctions biased using an operation voltage $V_{QCR}$. The probe resistor and the QCR are both equipped with an additional pair of current-biased NIS junctions. Using a calibration against the bath temperature (Supplementary Note 1 and Supplementary Fig. 1), the observed voltage excursions across these thermometer junctions provide us independent measures of the electron temperatures of the QCR, $T_{QCR}$, and of the probe resistor, $T_{probe}$. Figure 1e shows a control sample, which has additional superconducting wires in parallel with the QCR and probe resistors. The wires decouple the resistors from the electric currents associated with the resonator modes with no other significant effects. Thus, the difference between the behaviour of the active sample and that of the control sample can be attributed to microwave photons in the resonator.

The most important device parameters extracted from the experiments are listed in Table 1, including the length $L$ and the resonance frequency $f$ of the resonator. The probe resistance $R$ and its distance $x$ from the resonator edge determine the strength of the ohmic coupling between the probe electrons and the resonator mode. See Supplementary Note 2 for the relevance of these parameters in the thermal model used to describe the QCR.

**Principle of quantum-circuit refrigeration.** Let us consider sequential single-electron tunnelling through the NIS junctions of the QCR as described by Fermi's golden rule. Since the tunnelling process changes the electric charge in the resonator, the initial and final states of the resonator before and after an electron tunnelling event may be different[35]. As we discuss in Supplementary Note 3, the probability of this type of photon-assisted tunnelling is essentially proportional to an environmental parameter $\rho = \pi/(CR_K\omega_0)$, where $C$ is the effective capacitance of the resonator, $R_K$ is the von Klitzing constant and $\omega_0 = 2\pi f_0$.

In addition to a finite environmental parameter, the tunnelling events need to be energetically possible. Owing to the energy cost of $\Delta$ for an unpaired electron to enter a superconductor, there is essentially no tunnelling at vanishing operation voltages and low-electron temperatures. Thus, when the QCR is in this off state $V_{QCR} = 0$, the effect on the resonator is minimal.

Figure 2a shows an energy diagram for different tunnelling processes at the QCR with a finite operation voltage. Here electrons at various energy levels can overcome the energy cost $\Delta$ if they obtain an energy packet of $\hbar\omega_0$ from the resonator photon and of $eV_{QCR}/2$ from the bias voltage source. However, elastic tunnelling and photon emission are still exponentially suppressed by the lack of high-energy thermal excitations. Thus, this bias voltage regime, $\Delta - \hbar\omega_0 \lesssim eV_{QCR}/2 \lesssim \Delta$, is suitable for refrigerating the resonator.

In addition, the electron tunnelling current corresponding to photon-assisted tunnelling depends heavily on the state of the resonator. However, suppressed cooling power owing to low-resonator temperature does not imply that the QCR fails to carry its purpose—a ground-state system does not need to be initialized. Thus, instead of the cooling power, the important quantities describing the performance of the QCR are the relaxation and excitation rates that it introduces between the quantum levels of the resonator. See Supplementary Notes 5–8 for further discussion.

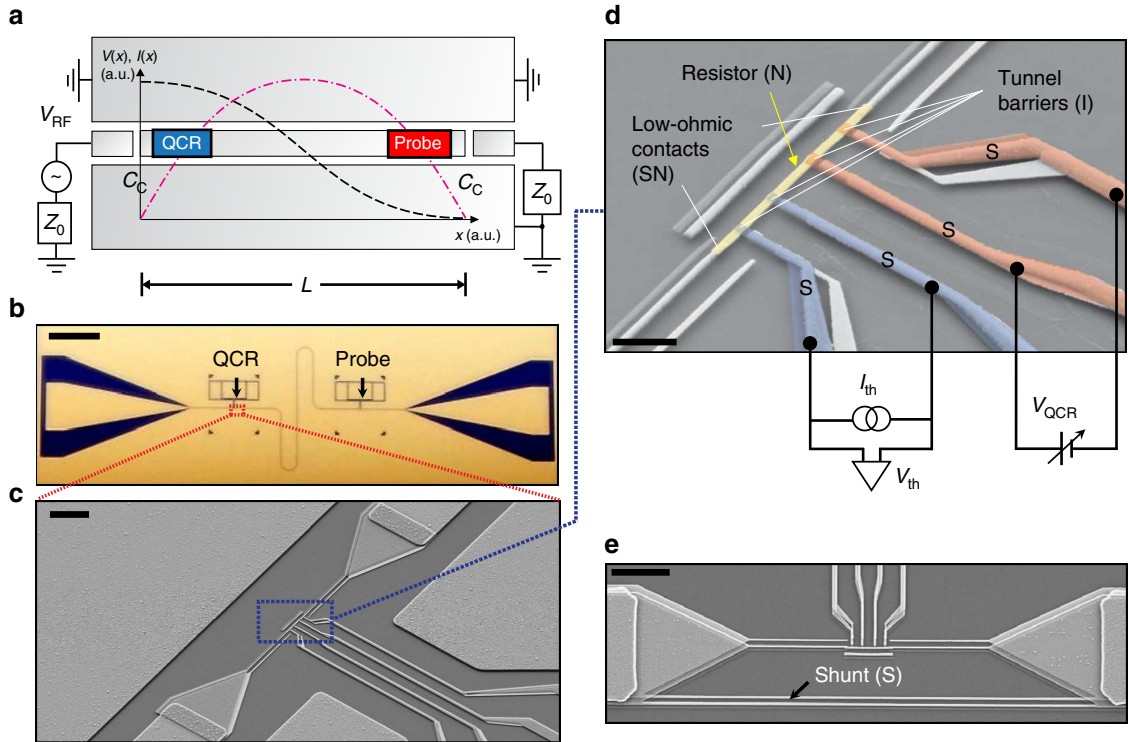

**Figure 1 | Experimental sample and measurement scheme.** (**a**) Schematic illustration of the active sample (not to scale) composed of a coplanar-waveguide resonator of length $L$ with an embedded QCR and probe resistor. The voltage ($V$, black dashed line) and current ($I$, red dashed dotted line) profiles of the fundamental resonator mode are shown together with the possibility to apply an external microwave drive, $V_{RF}$, to the resonator through a coupling capacitor $C_c$. The characteristic impedance of the microwave line is $Z_0 = 50\,\Omega$. (**b**) Optical micrograph of an active sample corresponding to **a**. The QCR and the probe resistor are indicated by the arrows. The large triangle-like features near the left and right ends of the chip are bonding pads for the microwave drive. The thin meandering feature connecting the QCR and the probe is the resonator. (**c**) Scanning electron microscope (SEM) image in the vicinity of the QCR. The centre conductor of the resonator is galvanically connected to aluminium leads, which are again connected to the normal metal of the QCR (centre of figure). (**d**) Coloured SEM image of the QCR with normal-metal (N), insulator (I) and superconductor (S) materials indicated. The refrigerator is operated in voltage bias, $V_{QCR}$, while the electron temperature of the normal metal is obtained from the voltage $V_{th}$ across a pair of NIS junctions biased with current $I_{th}$. (**e**) SEM image of the shunted control sample in the vicinity of the QCR. Scale bars, 1 mm (**b**), 5 μm (**c**), 1 μm (**d**) and 5 μm (**e**).

**Table 1 | Key device parameters.**

| Parameter | Symbol | Value | Unit |
|---|---|---|---|
| Resonator length | $L$ | 6.833 | mm |
| Fundamental resonance frequency | $f_0$ | 9.32 | GHz |
| Resistance of QCR and probe resistors | $R$ | 46 | $\Omega$ |
| Distance of the resistors from resonator edge | $x$ | 100 | μm |

These most important device parameters are extracted from the discussed experiments. The full list of parameters used in the thermal model can be found in Supplementary Table 1.

Using the parameters realized in our experiments, Fig. 2b and Supplementary Fig. 3 show theoretically obtained (Supplementary Note 3) rates for exciting the ground state of the fundamental resonator mode, $\Gamma^{T}_{0\to1}$, and for the relaxation of the first excited state, $\Gamma^{T}_{1\to0}$, as functions of the operation voltage. These rates are proportional to the environmental parameter and to the normal-state conductance of the tunnel junction, which are fixed in fabrication to obtain a desirable overall level for the rates. Importantly, the above-discussed energy constraints determine the behaviour of the rates as functions of the operation voltage.

We observe in Fig. 2b that both rates achieve their minima at low-operation voltages as expected for the off state of the QCR. Here the rates are roughly $10^4\,\mathrm{s}^{-1}$ and below which implies that

they do not dominate the relaxation dynamics of a quantum device with a natural lifetime below 100 μs. Slightly below the gap voltage, we have $1/\Gamma^{T}_{1\to0} < 10\,\mathrm{ns}$, which allows for quick initialization when desired. Here the excited-state population of the resonator solely corresponding to these rates is $\Gamma^{T}_{0\to1}/(\Gamma^{T}_{1\to0}+\Gamma^{T}_{0\to1}) < 10^{-3}$. If demonstrated in a superconducting qubit, these numbers would represent a clear improvement in the present state-of-the-art initialization schemes[14–20]. Interestingly, Fig. 2b also shows that somewhat below the gap voltage, photon-assisted tunnelling tends to drive the resonator to much smaller excited-state populations than the QCR electron system.

As discussed below, the experimentally achieved photon number of the resonator also depends on other heat conduction channels. However, note that the experimental parameters used to obtain the results from Fig. 2b onwards are chosen to conveniently reveal the QCR operation and are not optimal for high-finesse or low-temperature cooling. Optimization of the parameters is discussed in Supplementary Note 4.

**Observation of quantum-circuit refrigeration.** Figure 3a shows the measured changes in the electron temperatures of the QCR and of the probe resistor as functions of the QCR operation voltage. Slightly below the gap voltage $2\Delta/e$, both electron temperatures are significantly decreased. Here the high-energy electrons at the QCR overcome the superconductor energy gap

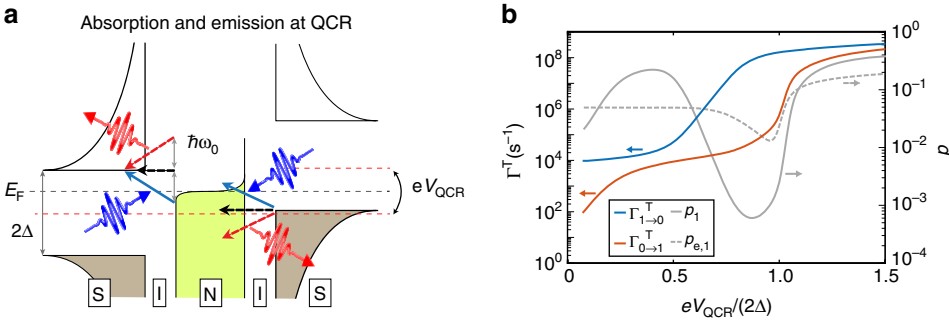

**Figure 2 | Electron tunnelling events and the resulting resonator transition rates. (a)** Schematic energy diagram illustrating sequential single-electron tunnelling events in the QCR: elastic tunnelling (black arrows), photon absorption from the resonator (blue arrows) and photon emission to the resonator (red arrows). The beginning of a straight arrow denotes the initial energy (vertical axis) and location (S or N) of the electron that tunnels, and the head of the arrow points to the final energy and location. The wavy arrows illustrate the photons absorbed (blue) and emitted (red) by the electrons. The coloured regions denote the occupied states and the white regions correspond to empty or absent states. The absence of states occurs at the insulators (I) and in the energy gap of size $2\Delta$ at the superconductor. The dashed arrows indicate tunnelling processes which are forbidden owing to no occupancy at the initial energy level or no vacancy at the final level. Although tunnelling takes place at a multitude of different energy levels, we only show the most likely events of each kind which either initiate or terminate at the edge of the superconductor energy gap where the density of states (black solid line at S) ideally diverges. The dashed lines show the Fermi levels of the normal metal (black, $E_F$) and of the superconductors (red). The work done by the voltage source in the course of an electron tunnelling event through the whole structure, $eV_{QCR}$, is depicted as corresponding shifts of the Fermi levels. **(b)** Relaxation ($\Gamma^T_{1\to0}$, blue line) and excitation ($\Gamma^T_{0\to1}$, red line) rates of the fundamental mode of the resonator and the corresponding steady-state excitation probability $p_1 = \Gamma^T_{0\to1}/\left(\Gamma^T_{0\to1} + \Gamma^T_{1\to0}\right)$ (grey line) as functions of the QCR operation voltage, $V_{QCR}$. The rates are calculated from the $P(E)$ theory (Supplementary Note 3) using the experimentally realized parameter values for the active sample and the two-state approximation. Supplementary Fig. 3 shows corresponding results for optimized parameters. The grey dashed line shows the excited-state population assuming that the resonator temperature equals that of the normal-metal electrons of the QCR.

and tunnel out of the normal metal, thus evaporatively cooling it. Typically, the observed temperature drop at the probe resistor would be simply explained by conduction of heat from it to the QCR electrons. However, this explanation is excluded by our observation that at operation voltages slightly above the gap voltage, the electron temperature in the QCR is elevated but the probe resistor remains cooled.

To explain the cooling of the probe resistor in Fig. 3a, we show a schematic energy diagram for the different types of tunnelling processes in Fig. 3b. In contrast to Fig. 2a, the operation voltage is above $2\Delta/e$, and hence elastic tunnelling is energetically favourable. In fact, elastic tunnelling is dominating here since photon-assisted tunnelling is suppressed by the small environmental parameter $\rho = 4.7 \times 10^{-3}$. At this operation voltage, elastic tunnelling mostly removes electrons below the Fermi energy of the normal metal and adds electrons above it, leading to the heating of the QCR electrons. However, elastic tunnelling has no direct effect on the resonator mode, which is predominantly cooled owing to photon absorption by the tunnelling electrons. Note that the emission of photons from the tunnelling electrons is suppressed by the relatively low-thermal population of the high-energy excitations in the normal metal.

In the highlighted region of Fig. 3a, the refrigeration of the resonator by photon-assisted tunnelling is competing with the power flowing into the resonator from the increasing electron temperature at the QCR. Since the QCR is located very close to the edge of the resonator where the mode current vanishes (Fig. 1a), the QCR electron system is rather weakly coupled to the resonator mode through ohmic losses. Thus, the dramatic rise of the electron temperature in the QCR at voltage $V_{QCR} \approx 2\Delta/e$ does not induce major changes in the temperature of the resonator or of the probe resistor. In an optimized design, the QCR can be placed at the very end of the resonator rendering the ohmic coupling negligibly weak. See Supplementary Note 4, Supplementary Fig. 2 and Supplementary Table 2 for details of the optimized design.

**Thermal model**. To analyse quantitatively the observed temperature changes in Fig. 3a, we introduce a thermal model shown in Fig. 3c. We theoretically model the photon-assisted tunnelling using the so-called $P(E)$ theory[35] for NIS tunnel junctions[36]. The dominating energy flows into the conduction electrons of the probe resistor are obtained from their coupling to the substrate phonons and to the resonator. The coupling to the resonator arises from ohmic losses in the resistor due to the electric current associated with the resonator photons[37]. Similar weak ohmic losses take place at the QCR. In addition, we take into account a weak residual heating of the probe resistor due to the power dissipation at the QCR, a constant thermal conductance to an excess bath and a constant heating of the resonator attributed to photon leakage from the high-temperature stages of the cryostat. Further details of the thermal model including the employed parameter values are given in Supplementary Note 2.

**Quantum-circuit refrigeration explained by thermal model.** For a given operation voltage and measured electron temperature at the QCR, we solve the temperatures of the probe resistor and the resonator from the thermal model (Supplementary Note 2) such that the different power flows in Fig. 3c balance each other. The theoretical prediction for the probe temperature is in very good quantitative agreement with our experimental observations as demonstrated in Fig. 3a. Thus, we may estimate the resonator temperature using that obtained from the thermal model as shown in Fig. 3d together with the average photon number (see also Supplementary Fig. 4). We observe that the resonator mode can be efficiently cooled down using the QCR operation voltage.

Figure 3a also shows that a theoretical prediction lacking the contribution from photon-assisted tunnelling is in clear disagreement with the measurements. This is a strong indication that the cooling power of the QCR originates from the direct absorption of resonator photons in the course of electron tunnelling.

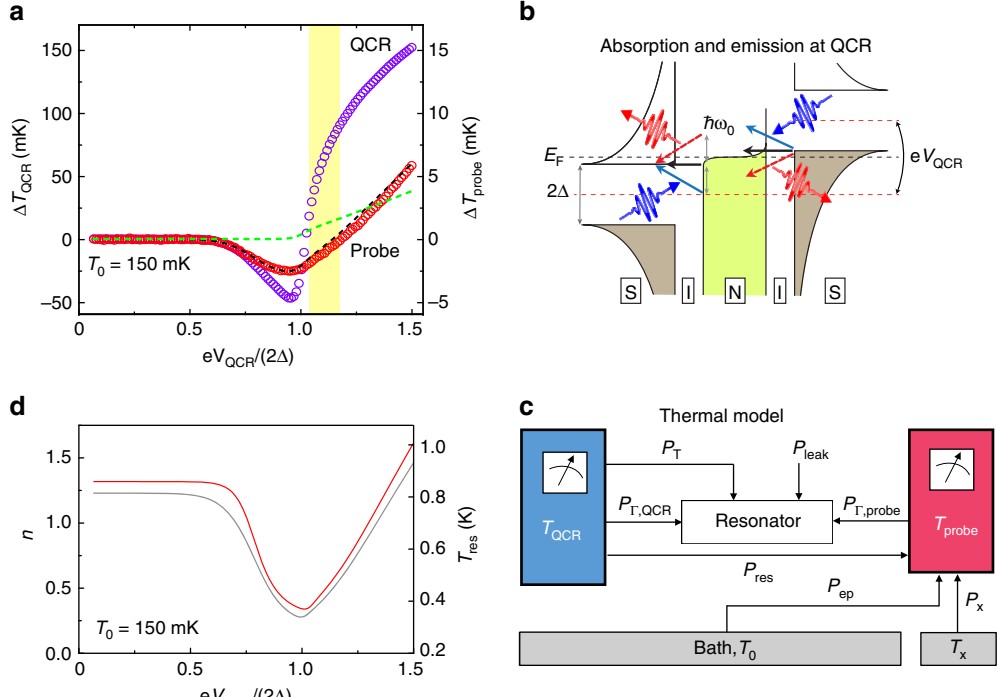

**Figure 3 | Quantum-circuit refrigeration and thermal model.** (**a**) Experimentally measured changes in the electron temperatures of the QCR, $\Delta T_{QCR}$ (purple circles) and of the probe resistor, $\Delta T_{probe}$ (red circles), as functions of the refrigerator operation voltage $V_{QCR}$. The dashed lines show the theoretical $\Delta T_{probe}$ with (black) and without (green) photon-assisted tunnelling. (**b**) Tunnelling diagram similar to that in Fig. 2a, but for a higher operation voltage corresponding to the region highlighted in yellow in **a**. Here only photon emission to the resonator is suppressed by the lack of thermal excitations. (**c**) Thermal model used for the experiment. Blue colour denotes the electron system of the QCR and red colour that of the probe resistor. Only the fundamental mode of the resonator is considered. The power $P_T$ arises from photon-assisted tunnelling; $P_{\Gamma,QCR}$ and $P_{\Gamma,probe}$ correspond to ohmic losses; $P_{ep}$ accounts for coupling between the probe electrons and the phonon bath at temperature $T_0$; $P_{res}$ denotes the residual heating power of the probe due to $V_{QCR}$; $P_{leak}$ accounts for leakage of photons to the resonator from high-temperature stages of the cryostat; and $P_x$ denotes excess power due to a constant thermal conductance $G_x$ to a reservoir at temperature $T_x$. Negative power implies the opposite direction of the energy flow with respect to the shown arrows. See Supplementary Note 2 for a detailed description of the model. (**d**) Resonator temperature ($T_{res}$, grey line) and average photon number ($n$, red line) solved from a thermal model corresponding to the measurements in **a**. See Supplementary Note 6 and Supplementary Fig. 4 for more information and for data at different bath temperatures.

**Effect of bath temperature on the refrigeration.** Figure 4a shows the temperature change of the probe resistor in a broad range of cryostat bath temperatures, $T_0$, and QCR operation voltages. A good quantitative agreement with the experimental data and the thermal model is obtained. For bath temperatures above 200 mK, the QCR operation voltage has a very weak effect on the probe resistor. This loss of probe sensitivity is explained by the quartic temperature dependence of the thermal conductance between the probe electrons and phonons given by supplementary equation (9) in Supplementary Note 1. The greater this thermal conductance is, the less sensitive the probe is to the changes of the resonator temperature.

**Comparison with a control sample.** Figure 4b shows results similar to those in Fig. 4a but obtained with the control sample, in which the ohmic losses at the QCR and at the probe due to the resonator modes are suppressed (Fig. 1e). Although residual heating is observed at high-operation voltages, there is no evidence of refrigeration at the probe. Thus, the cooling of the probe in the active sample must arise from the QCR acting on the resonator. This conclusion is also supported by Fig. 4c, where we show the maximum temperature drop of the probe for the two samples at various bath temperatures. Here the control sample exhibits no cooling and the theoretical prediction is in very good agreement with the experimental observations.

**Microwave response of the resonator.** To verify that the refrigerated resonator has a well-defined mode at the designed frequency, we introduce rf excitation to one of the input ports of the resonator as described in Fig. 1a. Although not necessary for the operation of the QCR, these ports are deliberately very weakly coupled to the mode, and hence there is essentially no transmission through the resonator. However, we study the resonance in Fig. 5 by measuring the electron temperatures of the QCR and of the probe resistor as functions of the frequency and power of the excitation. We observe a well-resolved resonance peak centred at $f_0 = \omega_0/(2\pi) = 9.32$ GHz in agreement with the design parameters (Supplementary Table 1). At the lowest probe powers, the electron temperature and the absorbed power are linearly dependent. Therefore, we can accurately extract the full-width at half-maximum, $\Delta f_0 = 70.8$ MHz, using a Lorentzian fit to the electron temperature. The obtained quality factor of the resonance, $Q = f_0/\Delta f_0 = 132$, indicates that the resonator supports a well-defined mode. Similar experiments on the control sample (Supplementary Fig. 5) yield a quality factor of 572 indicating that as expected, ohmic losses dominate in the active sample. We attribute the observed losses in the control sample to residual ohmic coupling owing to the finite impedance of the shunt.

**Discussion**

The main result of this work is the demonstrated principle that single-electron tunnelling between a normal metal and a

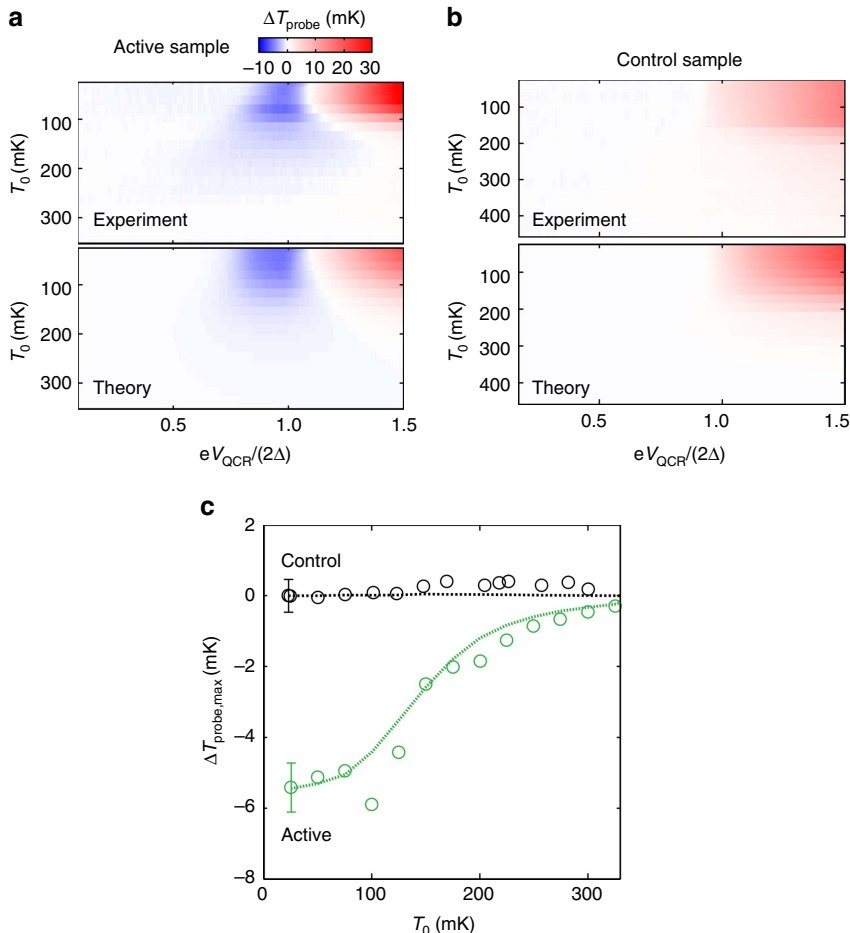

**Figure 4 | Comparison between the active sample and the control sample.** (**a**,**b**) Temperature change of the probe, $\Delta T_{\text{probe}}$, as a function of the QCR operation voltage and the bath temperature for experimental and simulated data in the case of the active sample (**a**) and the control sample (**b**). (**c**) Temperature changes of the probe from the active sample (green circles, extracted from **a**) and from the control sample (black circles, extracted from **b**) as functions of the bath temperature at the operation voltage corresponding to the maximum cooling point of the probe and the QCR, respectively. Dashed lines show the corresponding data obtained from the thermal model. The error bars indicate the maximum s.d. for each data set.

superconductor can be used to refrigerate a microwave resonator on demand. In the future, we aim to optimize the set-up for minimal ohmic and other spurious losses (Supplementary Notes 4 and 9) and for low-resonator temperatures by improving the design, nanofabrication, shielding and filtering. An ideally working QCR can potentially refrigerate a multitude of high-quality quantum circuits with many conceivable applications such as precise qubit initialization for large-scale, gate-based quantum computing, quantum-state engineering driven by dissipation[38] and active enhancement of ground-state population in quantum annealing[39,40]. When inactive, such an ideal refrigerator does not degrade the quantum coherence as desired. To decrease the base temperature achieved with our device, multiple refrigerators may be cascaded with each other[41] or with distant ohmic reservoirs at very low temperatures[42].

In our experiments, the relatively low-quality factor of the resonator is a result of the chosen measurement scheme: rather strong dissipative coupling between the resonator photons and the probe resistor is needed to achieve conveniently measurable temperature changes at the probe resistor when the resonator is being refrigerated. In the future, different probe schemes may be employed where such limitation is absent. These include amplification and analysis of the resonator output signal[43,44] and measurement of the resonator occupation using a superconducting qubit[8,45]. Furthermore, the ohmic losses due

to the QCR may be overcome by minimizing its resistance and by moving it to the end of the resonator where the current profile of the resonator mode ideally vanishes. Importantly, this does not reduce the cooling power of the QCR, which utilizes the voltage profile. As discussed in more detail in Supplementary Note 4, this type of straightforward improvements of the QCR are expected to allow for resonator quality factors[46] in the range of $10^6$ when the QCR is inactive. Such a level of dissipation is low enough for the undisturbed operation of quantum devices in their known applications.

## Methods

**Sample fabrication.** The samples are fabricated on four-inch prime-grade intrinsic silicon wafers with 300-nm thick thermally grown silicon dioxide. The resonator is defined with optical lithography and deposited using an electron beam evaporator, followed by a lift-off process. The evaporated metal film consists three layers from bottom to top: 200 nm of Al, 3 nm of Ti and 5 nm of Au. Here gold is used to prevent oxidation and titanium is introduced to avoid the diffusion of gold into the aluminium layer.

The nanostructures are defined by electron beam lithography. Here we employ a bilayer resist mask consisting of poly(methyl methacrylate) and poly[(methyl methacrylate)-co-(methacrylic acid)] to enable three-angle shadow evaporation. The trilayer nanostructures are deposited in an electron beam evaporator, with *in situ* oxidation in between the first layer (Al) and the second layer (Cu) to form the NIS tunnel junctions. The third layer (Al) forms a low-ohmic contact with the second layer, which functions as the normal metal in our low-temperature experiments. A lift-off process is performed to remove excess metal. See Table 1 for the resulting parameter values.

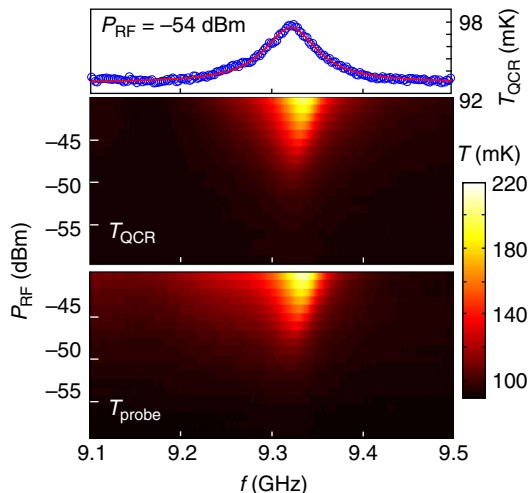

**Figure 5 | Observation of the fundamental resonance.** Experimentally observed electron temperatures of the QCR and of the probe resistor as functions of the frequency and power of the external microwave drive. The measurement scheme is illustrated in Fig. 1a. We also show a trace of the refrigerator temperature at $-54$-dBm room temperature power (markers) together with a Lorentzian fit (solid line). The indicated room temperature power levels decrease according to Supplementary Fig. 1c before reaching the sample.

Improvements to the fabrication process towards lowering the amount of electric losses in the resonator are discussed in Supplementary Note 4.

**Measurements.** For cryogenic electrical measurements, the sample holders are mounted to a cryogen-free dilution refrigerator with a base temperature of 10 mK. The silicon chip supporting the sample is attached with vacuum grease to the sample holder and wedge bonded to the electrical leads of the printed circuit board using aluminium wires. For each dc line, we employ an individual resistive Thermocoax cable that runs without interruption from the mixing chamber plate of the dilution refrigerator to room temperature.

The NIS thermometers are biased with floating battery-powered current sources and the voltage drops across these junctions are amplified with high-impedance battery-powered voltage preamplifiers before optoisolation and digitalization with an oscilloscope. In the experiments studying different QCR operation voltages, we sweep $V_{QCR}$ at a rate of $\sim 20\,\mu V\,s^{-1}$ using an output of an arbitrary function generator that is connected to the cryostat through an optoisolator. The sweep is repeated 10 times at each bath temperature.

In the rf measurements, the sinusoidal drive signal is generated by a variable-frequency microwave source and guided to the sample through low-loss coaxial cables, which are attenuated at different temperature stages of the cryostat as shown in Supplementary Fig. 1c.

See Supplementary Note 1 for the details of the NIS thermometry including calibration data.

**Modelling.** All numerical computations are carried out using regular desktop computers. See Supplementary Note 2 for a detailed description of the theoretical model used in this work.

**Data availability.** The data and codes that support the findings of this study are available from the corresponding authors on request.

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

## Acknowledgements

The authors thank M. Meschke, J.P. Pekola, M. Silveri and H. Grabert for insightful discussions. This work is supported by the European Research Council under Starting Independent Researcher Grant No. 278117 (SINGLEOUT) and under Consolidator Grant No. 681311 (QUESS), by the Academy of Finland through its Centres of Excellence Program (Grant Nos 251748 and 284621) and Grants (Nos 135794, 272806, 265675, 276528, 286215 and 305306), the Emil Altonen Foundation, the Jenny and Antti Wihuri Foundation and the Finnish Cultural Foundation. We also acknowledge the provision of facilities and technical support by Aalto University at Micronova Nanofabrication Centre.

## Author contributions

K.Y.T. fabricated the samples, developed and conducted the experiments, and analysed the data. M.P. contributed to the sample fabrication, measurements and data analysis. R.E.L. and J.G. contributed to the measurements. S.M. contributed to the theoretical analysis of the system. M.M. provided the initial ideas and suggestions for the experiment, and supervised the work in all respects. All authors discussed both experimental and theoretical results and commented on the manuscript, which was written by K.Y.T. and M.M.

## Additional information

**Competing interests:** The authors declare no competing financial interests.

**Publisher's note**: 

