## [Peer review file · Nature Communications]

Reviewers' comments:

Reviewer #1 (Remarks to the Author):

Report on "Quantum-Circuit Refrigerator" by K. Y. Tan et al.

The authors report direct cooling of a superconducting resonator mode through normal metal-insulator-superconductor junctions. They reveal this result by a decreased electron temperature at a probing resistor coupled to the resonator, even when electrons in the refrigerator are at an elevated temperature. These conclusions are supported by a control experiment, and also by a good quantitative agreement between a theoretical model and the experiment itself. Furthermore, these conclusions are confirmed in a wide range of biasing voltages and temperatures of the lattice phonons. The authors verify as well that the resonator possesses a well-defined resonance.

This experimental work here presented is interesting and well performed, and reports relevant results on the observation of photon-assisted cooling of a resonator mode. These results can definitely have impact on the research field of thermal transport in superconducting circuits at the nanoscale. The described setup can potentially act, in an optimized configuration, as a refrigerator for a multitude of quantum circuits, and can potentially have further applications as precise initialization of quantum bits, and in enhanced quantum information architectures.

The treated topic is timely, the manuscript is clearly written, the figures are clear and well presented, and the bibliography seems to have been properly selected.

The manuscript is suitable for the readership of Nature Communications, and could be published after the authors address the issues that are listed below:

- i) It would be very useful for the reader if the authors described shortly at the beginning of the paper the physical principle of cooling due to photon-assisted tunneling. Although described in the Supplementary Information, I believe it would be an added value for the paper.
- ii) It would be interesting if the authors could discuss about the achieved cooling power in their structure. What is the main limiting factor to it?
- iii) Could the authors give an estimate of the coefficient of performance (COP) of their cooler, and how does it compare with direct refrigerators realized with superconducting tunnel junctions?
- iv) Could the authors briefly discuss how to enhance cooling of the resonator? Is there something related as well to the materials combination?
- v) Could the authors discuss more in detail the possible impact of back-action of the cooler onto the resonator performance? How to limit the impact of such back-action on the general coherence characteristic of the system?

Reviewer #2 (Remarks to the Author):

This manuscript shows how a particular configuration of normal-insulator-superconductor tunnel junctions called a quantum circuit refrigerator (QCR) by the authors can be used to present a controllable electromagnetic load to a microwave resonator. When active, the QCR can be used to cool the resonator via photon-induced electron tunneling through the QCR where the resonator is the source of the photons. The experimental data is convincing, particularly the data highlighted in gold in Fig 2a where the QCR is heated by the applied voltage but the resonator is cooled. The data is strongly supported by theory presented in the SI whose physical basis is reasonable and whose predictions are in good agreement with the data.

The authors argue that the QCR could be used to assist state preparation in future quantum circuits and the results of Fig. 2d showing a large change in the average resonator photon number support this claim. In the experimental data presented, dissipation in the QCR significantly lowers the resonator quality factor which is an obstacle to the immediate application of the work. However, the authors make detailed and plausible arguments in the supplemental information that an optimized device would not experience this level of loss. Some referees might argue that this important point should be demonstrated before publication but I feel that such a requirement sets too high a bar. If all ideas had to be fully realized from conception to practical implementation before the work was published, journals would be slim indeed.

I have two trivial comments and one medium-size comment related to the clarity of the manuscript.

Trivial:

1. Fig. 1d. From context, I believe the red and blue leads are superconductors but this should probably be indicated in the figure.
2. Line 272 of the SI. There is a typographic error in the spelling of junction.

Medium:

1. The fundamental physical mechanism behind the QCR is that at certain voltage biases, the tunnel junctions of the QCR absorb more microwave photons from the resonator than they emit and that the DC power balance within the QCR is largely irrelevant to the behavior of the resonator. However, the manuscript fails to clearly explain why the voltages in the gold region of Fig. 2a (as well as slightly lower voltages) achieve this desirable condition. An effort at explanation is made in lines 100-102 of the main text but this material is not a successful explanation. Fig. 2b clearly is also an attempt at explanation, but I find this figure at best unhelpful and at worst possibly wrong. For example, the black arrows denoting elastic tunneling only show electrons leaving the central normal electrode which violates charge conservation. Also, the lower black arrow shows an electron tunneling into the forbidden region of the superconducting density of states. Also, the use of two black arrows at discrete energies is confusing because all the electrons in the normal electrode above the top of the left superconductor's gap are able to tunnel. Finally, the arrows showing photon-assisted tunneling might contain an explanation of why absorption is favored over emission based on the accessible final densities-of-states but this is just my own speculation. Of

course the detailed model in the supplemental information must contain the answer, but only the most determined of readers will be willing to evaluate on their own the tunneling expressions given in the SI. To maximize the clarity and impact of the manuscript, I encourage the authors to explain this central piece of their work more clearly.

In summary, I recommend this very interesting piece of work for publication subject to the stylistic recommendation above.

Reviewer #3 (Remarks to the Author):

The experimental set-up studied by the authors consists of two SINIS structures directly coupled by their normal-metal strips to the opposite ends of a microwave CPW resonator. When the refrigerator cools its normal metal strip, the thermometer registers a small (a few mK) drop in temperature. An unexpected finding is that the thermometer readings remain BELOW the “bath” temperature T_0 over a range of the refrigerator’s bias voltages that correspond to OVERHEATING of the refrigerator above T_0 . The authors attribute this unusual effect to the cooling of the resonator mode due to the photon-assisted tunneling in the SINIS refrigerator (which remains colder than T_0 even at moderate overheating of the refrigerator), and cooling of the thermometer due coupling to this “cold” resonance mode.

Even if the authors’ interpretation of the observed effect is correct, it is unlikely that the effect can find applications in quantum computing, as the authors claim in the introduction. Let’s estimate the MAXIMUM power P_{max} that could flow from the thermometer into the resonator mode. The lifetime τ of photons in the studied resonator is about 1 ns ($\tau \approx Q/\omega_r$, where $Q \sim 100$ is the quality factor of the resonator, ω_r is its resonance frequency). Assuming that the occupation number for photons in this mode does not exceed 1, $P_{\text{max}} \approx \hbar\omega_r/\tau$ is limited to $\sim 10^{-14}$ W. This estimate assumes that all photons are absorbed in the refrigerator due to the photon-assisted tunneling, and there is no photon “backflow” from the overheated refrigerator, which, of course, is not the case. No wonder that with such a tiny cooling power, the observed T drop did not exceed 5 mK.

The P_{max} values cannot be improved much. Indeed, most solid-state qubits, both superconducting and semiconducting, operate at the base temperature of dilution refrigerators, $T \approx 20\text{-}50\text{mK}$. The authors have not demonstrated that the SINIS refrigerator can depopulate photons in the resonator mode at such a low temperature. Also, with reduction of active losses in the resonator (a must for quantum-computing applications), P_{max} should decrease even more - $P_{\text{max}} \propto 1/Q$. This could be one of the reasons why the authors observed no cooling in the test device where both the refrigerator and thermometer were “shunted” by superconducting wires. The Q factor in the test device (~ 1000 , Fig. S5) was about 10 times greater than that for the studied device which demonstrated cooling ($Q \sim 100$, Fig.4).

In conclusion, it is unlikely that this method will find applications in quantum computing, and the claims that the authors make in the introduction and conclusion are far-fetched. For this reason, the manuscript cannot be recommended for publication in Nature Communications.

Detailed response to the report of Reviewer 1

We thank the reviewer for finding our work interesting and well-performed, and for recommending publication in Nature Communications. Below, we provide our point-by-point response to the suggestions for changes pointed out by the reviewer. We have modified the manuscript accordingly.

Reviewer: The manuscript is suitable for the readership of Nature Communications, and could be published after the authors address the issues that are listed below:

i) It would be very useful for the reader if the authors described shortly at the beginning of the paper the physical principle of cooling due to photon-assisted tunneling. Although described in the Supplementary Information, I believe it would be an added value for the paper.

Response: We thank the reviewer for pointing this out. We have rearranged the old Fig. 2 and added a new figure (Fig. 2) to address this point. We have also added almost a full paragraph to the introduction and a full new section just after the introduction to make physical principle of cooling with photon-assisted tunneling clear. Note that in the original submission, the figure diagrammatically describing photon-assisted tunneling had been corrupted, and consequently rendered the understanding of the principle challenging.

Reviewer: ii) It would be interesting if the authors could discuss about the achieved cooling power in their structure. What is the main limiting factor to it?

Response: We have added a brief discussion in the main text and a new full section in the Supporting Information about the cooling power of the QCR. We also note that the cooling power is not the main figure of merit of a QCR since it heavily depends on the quantum state of the device that is being cooled. Instead, the transition rates that the QCR induces on the quantum device provide the useful information needed to estimate the effect of the refrigeration. To better convey this message, we have added a figure showing the induced transition rates (Fig. 2b in the revised manuscript) and added corresponding information in the main text.

Reviewer: iii) Could the authors give an estimate of the coefficient of performance (COP) of their cooler, and how does it compare with direct refrigerators realized with superconducting tunnel junctions?

Response: In the revised manuscript, we discuss the coefficient of performance in a full new section in the Supporting Information. We give a rough analytical upper bound for it in good operation conditions. We also compare the coefficient of performance of the QCR to that of the usual NIS cooler and highlight their qualitative differences.

Reviewer: iv) Could the authors briefly discuss how to enhance cooling of the resonator? Is it there something related as well to the materials combination?

Response: We have added a brief discussion about the cooling power to the main text and a full new section to the Supporting Information, but since the cooling power is not the main figure of merit of a

QCR, but the transition rates are, we do not see it necessary to extensively describe its optimization. However, let us write here how one could increase the cooling power. The cooling of the resonator is proportional to the environmental parameter $\rho = \frac{\pi}{CR_K\omega_0}$ [Supplementary Information equation (S15)] and to the thermal population of the resonator. Assuming that the resonator photon temperature remains constant and the resonator capacitance dominates the total effective capacitance of the environment, C , then $\rho \propto \frac{1}{\sqrt{C}}$. Thus, reducing the resonator capacitance will enhance the cooling. In addition, the QCR cooling power is directly proportional to the QCR bias current, I_{QCR} , if the other parameters remain constant. Hence the cooling power can be enhanced by lowering the tunnel resistance $R_T \propto \frac{1}{I_{\text{QCR}}}$ at least as far as it remains in the sequential-tunneling regime. Ideally, there should be no material-dependent parameters in the photon absorption and emission rates (Eq. S19), and hence using different materials is not expected to significantly enhance the resonator cooling.

Reviewer: v) Could the authors discuss more in detail the possible impact of back-action of the cooler onto the resonator performance? How to limit the impact of such back-action on the general coherence characteristic of the system?

Response: In the revised manuscript, we highlight the importance of the transition rates induced on the resonator by the QCR. To this end, we added Fig. 2b. The figure shows that at low operation voltages, the lifetime induced by the back action is much longer than the $\sim 100\text{-}\mu\text{s}$ natural lifetime of the present state-of-the-art planar superconducting qubits. Furthermore, the optimized design analyzed in Fig. S3 exhibits orders of magnitude weaker back action.

Detailed response to the report of Reviewer 2

We thank the reviewer for stating that our data and analysis is solid, for recommending publication, and for considering that our results and analysis support our claim that the QCR may be used to assist state preparation in future quantum circuits. Below, we provide our point-by-point response to the stylistic recommendations pointed out by the reviewer. We have revised the manuscript accordingly.

Reviewer: I have two trivial comments and one medium-size comment related to the clarity of the manuscript.

Trivial:

1. Fig. 1d. From context, I believe the red and blue leads are superconductors but this should probably be indicated in the figure.

Response: We have added in Fig. 1d characters 'S' on top of the leads to resolve this issue.

Reviewer: 2. Line 272 of the SI. There is a typographic error in the spelling of junction.

Response: The typo has been corrected.

Reviewer: Medium:

1. The fundamental physical mechanism behind the QCR is that at certain voltage biases, the tunnel junctions of the QCR absorb more microwave photons from the resonator than they emit and that the DC power balance within the QCR is largely irrelevant to the behavior of the resonator. However, the manuscript fails to clearly explain why the voltages in the gold region of Fig. 2a (as well as slightly lower voltages) achieve this desirable condition. An effort at explanation is made in lines 100-102 of the main text but this material is not a successful explanation. Fig. 2b clearly is also an attempt at explanation, but I find this figure at best unhelpful and at worst possibly wrong. For example, the black arrows denoting elastic tunneling only show electrons leaving the central normal electrode which violates charge conservation. Also, the lower black arrow shows an electron tunneling into the forbidden region of the superconducting density of states. Also, the use of two black arrows at discrete energies is confusing because all the electrons in the normal electrode above the top of the left superconductor's gap are able to tunnel. Finally, the arrows showing photon-assisted tunneling might contain an explanation of why absorption is favored over emission based on the accessible final densities-of-states but this is just my own speculation. Of course the detailed model in the supplemental information must contain the answer, but only the most determined of readers will be willing to evaluate on their own the tunneling expressions given in the SI. To maximize the clarity and impact of the manuscript, I encourage the authors to explain this central piece of their work more clearly.

We thank Reviewer 2 for pointing out this error. Figure 2b was indeed corrupted in the originally submitted manuscript, and consequently the right set of arrows was shifted left. The top right black arrow is originally intended to visualize elastic tunneling from the right superconductor to the normal metal, not from the normal metal to the left superconductor. We have corrected this error in the revised manuscript (Fig 2a) and extended the caption. In addition, we have added Fig. 2b that describes the photon-assisted transition rates and significantly extended the related discussion in the main text: we added a full new

section discussing the operation principle and rewrote the section discussing the experimental observation of quantum-circuit refrigeration. We hope that these changes render the manuscript main text successful in describing the central operation principle of the QCR.

Reviewer: In summary, I recommend this very interesting piece of work for publication subject to the stylistic recommendation above.

Response: We are grateful to Reviewer 2 for his/her interest in our work.

Detailed response to the report of Reviewer 3

We thank the reviewer for his/her comments. The reviewer questions the potential applicability of our QCR in quantum computing, but as we describe in our point-by-point response below, this may be due to a misunderstanding. Nevertheless, we have removed our claims on the applicability and revised the manuscript such that it provides a clearer message what is the function of the QCR.

Reviewer: Even if the authors' interpretation of the observed effect is correct, it is unlikely that the effect can find applications in quantum computing, as the authors claim in the introduction.

Response: We presented no claims in the introduction section of the original submission about the future applications of our work in quantum computing. According to the journal guidelines however, we discuss potential applications in the abstract and discussion section. Although we feel this discussion about the future was in the originally submitted manuscript already careful enough to be scientifically sound, we have taken out all claims about the applicability of our work and replaced them by stating that in general, a well-working quantum refrigerator has potential applications. Consequently, we think that we have fully addressed to the criticism of the reviewer regarding potential applications.

Reviewer: Let's estimate the MAXIMUM power P_{\max} that could flow from the thermometer into the resonator mode. The lifetime τ of photons in the studied resonator is about 1 ns ($\tau \approx Q/\omega_r$, where $Q \sim 100$ is the quality factor of the resonator, ω_r is its resonance frequency). Assuming that the occupation number for photons in this mode does not exceed 1, $P_{\max} \approx \hbar\omega_r/\tau$ is limited to $\sim 10^{(-14)}$ W. This estimate assumes that all photons are absorbed in the refrigerator due to the photon-assisted tunneling, and there is no photon "backflow" from the overheated refrigerator, which, of course, is not the case. No wonder that with such a tiny cooling power, the observed T drop did not exceed 5 mK.

Response: Here, the reviewer gives arguments why the cooling power from the thermometer to the resonator mode, P_{\max} , is low. However, this quantity is not important for the actual operation of the QCR, but it is just related to the way we chose to probe the resonator temperature in our experiments. As we point out in the conclusions, there are also several other ways to probe the temperature such that no issues with low heat transport arise. In fact, ideally no power is exchanged if the temperature is measured using a dispersively-coupled qubit.

Since the reviewer considers P_{\max} as an important quantity in relation to potential applications of the QCR, it is possible that the reviewer has misunderstood the concept of the QCR. Thus let us state that the key advantage of the QCR is that it can directly cool under-damped quantum systems such as the resonator in our case. We do not intent to present it as a cooler for over-damped systems such as that of the thermometer electrons. We have revised the main text and figures of the manuscript to stress this point: we have added a full new section describing the operation principle of the QCR. We also stress that even the cooling power from the resonator to the QCR is not a good figure of merit for the QCR since it heavily depends on the quantum state of the resonator. Instead, one should focus on the transition rates that the QCR induces on the resonator. These rates are shown in Fig. 2b of the revised manuscript.

Reviewer: The P_{\max} values cannot be improved much. Indeed, most solid-state qubits, both superconducting and semiconducting, operate at the base temperature of dilution refrigerators, $T \approx 20$ -

50mK. The authors have not demonstrated that the SINIS refrigerator can depopulate photons in the resonator mode at such a low temperature. Also, with reduction of active losses in the resonator (a must for quantum-computing applications), P_{\max} should decrease even more - $P_{\max} \propto 1/Q$. This could be one of the reasons why the authors observed no cooling in the test device where both the refrigerator and thermometer were “shunted” by superconducting wires. The Q factor in the test device (~ 1000 , Fig. S5) was about 10 times greater than that for the studied device which demonstrated cooling ($Q \sim 100$, Fig.4).

Response: The reviewer is concerned that the power from the thermometer to the resonator, P_{\max} , is reduced when the quality factor of the quantum device is increased. This just shows that the approach that we have used for thermometry in this work cannot be used for devices with very high quality factors. We already thoroughly discussed this issue in the original submission and showed in the Supplementary Information (section “Minimizing undesired losses due to the QCR”) that the requirement of high quality factors does not pose a problem for the operation nor the integration of the QCR with quantum devices.

The reviewer also writes that we have not demonstrated the operation at very low temperatures. This is true, but we already pointed out in the conclusions of the original submission that we leave this for future research. The main reason for this is the above-discussed fact that at low temperatures, P_{\max} is greatly reduced, and hence we need a different type of a thermometer. However, we and the two other reviewers consider that the first demonstration of a QCR we present is important enough that the work should be published. Quantum technology is a rapidly developing field and new ideas such as this should be communicated to the community in a timely manner. The development of a different type of a thermometer would mean a delay of the order of a year. A further reason why the demonstration of the QCR at lower temperatures is not considered critical at this stage is that the QCR is actually expected to work better at lower temperatures as we illustrate in Fig. S3.

Reviewer: In conclusion, it is unlikely that this method will find applications in quantum computing, and the claims that the authors make in the introduction and conclusion are far-fetched. For this reason, the manuscript cannot be recommended for publication in Nature Communications.

Response: We have shown above that the main concern of the reviewer about the low power from the thermometer to the resonator has no effect on the possible applicability of our work. Furthermore, we have removed all claims from the revised manuscript about the future applicability of our work and replaced them with general statements about applications of quantum refrigerators. Thus we hope that we have satisfied the reviewer and that our work can be published in Nature Communications.

REVIEWERS' COMMENTS:

Reviewer #1 (Remarks to the Author):

The authors have provided answers to my comments in an exhaustive and detailed way. Furthermore, they have improved their manuscript in several parts so to make the paper, in general, more readable and understandable for the scientific readership.

Also looking at the answers to the questions and comments raised by the other reviewers it seems that the authors provided convincing motivations, and useful modifications to the text of the paper.

Concerning Reviewer #3 point of view, to my mind the authors motivated clearly to her/him what they intended to show, and what they did demonstrate in their experiment. Her/his judgment against publication in Nature Communications seems more originated by obstinacy to the paper than something else. Moreover, the reasons she/he raises to support this decision are, from my point of view, just a matter of opinion.

To conclude, the paper is now suitable for publication in Nature Communications in the present form.

Reviewer #3 (Remarks to the Author):

The current version of the manuscript is much improved, and I recommend it for publication in Nature Communications.